# VID-LLM: A COMPACT VIDEO-BASED 3D MULTIMODAL LLM WITH RECONSTRUCTION–REASONING SYNERGY

**Haijier Chen[1*], Bo Xu[1,2*], Shoujian Zhang[1†], Haoze Liu[1], Jiaxuan Lin[1], Jingrong Wang[3]**

[1] School of Geodesy and Geomatics, Wuhan University
[2] Hubei Luojia Laboratory
[3] School of Architecture and Urban Planning, Shenzhen University

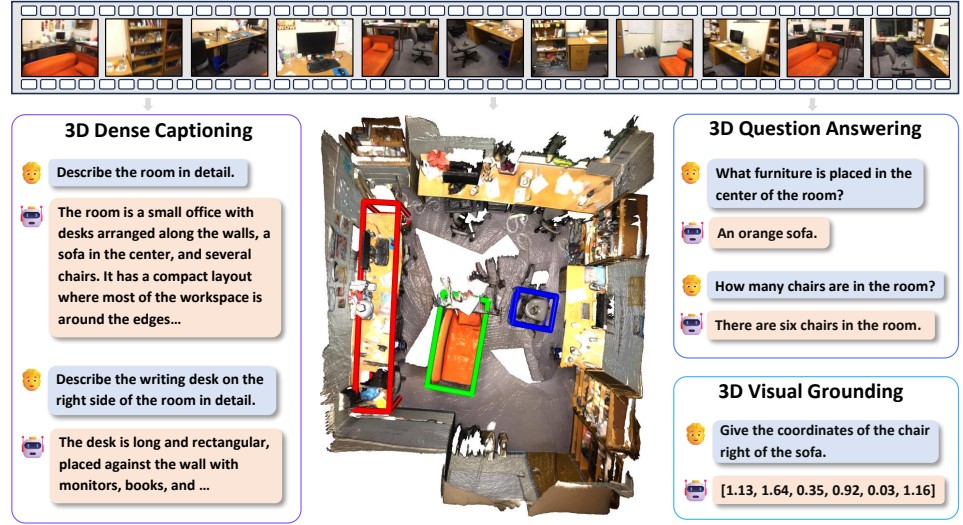

Figure 1: We propose **Vid-LLM** to achieve diverse 3D vision-language reasoning tasks using only video inputs.

## ABSTRACT

Recent developments in Multimodal Large Language Models (MLLMs) have significantly improved Vision–Language (VL) reasoning in 2D domains. However, extending these capabilities to 3D scene understanding remains a major challenge. Existing 3D Multimodal Large Language Models (3D-MLLMs) often depend on 3D data inputs, which limits scalability and generalization. To address this limitation, we propose Vid-LLM, a video-based 3D-MLLM that directly processes video inputs without requiring external 3D data, making it practical for real-world deployment. In our method, the geometric prior are directly used to improve the performance of the sceen perception. To integrate the geometric cues into the MLLM compactly, we design a Cross-Task Adapter (CTA) module to align the 3D geometric priors with the vision-language representations. To ensure geometric consistency and integrity, we introduce a Metric Depth Model that recovers real-scale geometry from the reconstruction outputs. Finally, the model is fine-tuned with a two-stage distillation optimization strategy, realizing fast convergence and stabilizes training. Extensive experiments across diverse benchmarks verified the effectiveness of our method on 3D Question Answering, 3D Dense Captioning and 3D Visual Grounding tasks, demonstrating the superior multi-task capabilities. Project page: https://chenhaijier.github.io/Vid-LLM/.

---

* Equal contribution.
† Corresponding author: shjzhang@sgg.whu.edu.cn

# 1 INTRODUCTION

Recent advances in Large Language Models (LLMs) (Vaswani et al., 2017; Radford et al., 2019; Naveed et al., 2025) and Multimodal Large Language Models (MLLMs) (Zhang et al., 2024a; Yin et al., 2024; Wu et al., 2023) have reinforced the paradigm of language as a universal interface, substantially improving cross-modal perception and reasoning. Extending this progress to 3D, recent research has focused on 3D-aware Multimodal Large Language Models (3D-MLLMs) (Ren et al., 2025), which unify 3D scene understanding and vision–language reasoning under a linguistic interface. This line of work underscores the importance of grounding language in persistent 3D spatial representations (Cheng et al., 2024a; Roh et al., 2022), offering a unified pathway toward systematic scene-level reasoning.

Recent studies have made substantial progress in 3D vision–language (3D VL) reasoning (Chen et al., 2024c; Huang et al., 2023b), yet most approaches rely on complex 3D inputs, incurring high costs in data collection, preprocessing, and computation. Some models rely on point clouds or reconstructed scenes augmented with rendered views or semantic–geometric features (Hong et al., 2023a; Fu et al., 2024), while others adopt simpler inputs but still depend on explicit 3D scene representations such as reconstructed objects aligned with semantic representations (Chu et al., 2024; Huang et al., 2023a; 2024). Despite their effectiveness, these pipelines depend on depth, poses, or external modules, leading to substantial data and engineering overhead as well as high memory and latency costs. This rigid input requirements and system complexity fundamentally limit the scalability and transferability of current 3D-MLLMs.

To overcome these limitations, a more general solution is to enable the model to directly reconstruct scene geometry from video (Leroy et al., 2024; Wang et al., 2024), thereby eliminating the reliance on external depth, pose, or registration modules. More importantly, reconstruction and reasoning are intrinsically interdependent: geometric structures underpin semantic understanding, while semantic reasoning, in turn, provides contextual priors that guide and refine geometric modeling (Cheng et al., 2024a; Ha & Song, 2022).

In this work, we introduce Vid-LLM, a compact model that jointly performs reconstruction and 3D vision–language reasoning from monocular video inputs, as illustrated in Fig. 1. The core component of Vid-LLM is a Cross-Task Adapter (CTA) that tightly couples reconstruction with reasoning, enabling intrinsic geometry–semantics interaction with mutual reinforcement and constraint. CTA disentangles geometry-aware and language-aware features; the geometric stream is then processed by a Global-Frame Attention backbone and specialized heads to estimate camera poses and relative depth, followed by a Metric Depth Model for real-scale calibration. The recovered 3D information is then fused with semantic features to construct 3D patches, which are fed into the LLM for spatial reasoning. Finally, a two-stage training strategy ensures convergence and improves overall performance. Extensive experiments across diverse 3D vision–language benchmarks demonstrate the performance of Vid-LLM and confirm its effectiveness as a practical and scalable framework for video-based 3D multimodal reasoning.

Our main contributions are summarized as follows:

- We propose Vid-LLM for versatile 3D scene understanding. The framework does not rely on dense 3D inputs or prior poses, making it practical for real-world deployment.

- We design a Cross-Task Adapter to align the 3D geometry priors with VL representations, boosting the integration of 3D visual geometry priors into MLLM. A two-stage training strategy is further adopted to improve the stability and performance.

- Extensive experimental evaluations are conducted on real datasets to evaluate the performance of our method. The experimental results demonstrate that our method achieves superior performance in terms of question answering, dense captioning and visual grounding. We will publish our code to facilitate communication.

# 2 RELATED WORK

3D-MLLMs have achieved significant advances in 3D scene understanding, yet their reliance on explicit 3D data still limits scalability and applicability. Meanwhile, progress in 3D reconstruction

shows that geometry can be directly reconstructed from videos. Integrating such geometric priors into 3D-MLLMs represents a promising approach to enhance semantic grounding. We therefore review related work in three directions: (i) 3D-MLLMs, (ii) 3D reconstruction, and (iii) geometry priors in vision-language models.

**3D-aware Multimodal Large Language Models (3D-MLLMs).** 3D-MLLMs aim to unify 3D scene understanding and vision–language reasoning within a unified linguistic interface, representing an important extension of multimodal large language models (MLLMs). Existing approaches predominantly rely on explicit geometric inputs: some leverage point clouds or reconstructed scenes, often augmented with rendered views, region-level alignments, or condensed 3D feature grids to support large-scale embodied training (Hong et al., 2023a; Chen et al., 2024b;c; Fu et al., 2024; Huang et al., 2023b); others map 3D features to the language space and model spatial relations to enable interactive dialogue (Huang et al., 2023a; 2024; Zheng et al., 2025b; Huang et al., 2025). Despite their progress, these methods invariably depend on complex inputs such as point clouds, reconstructed scenes, multi-view renderings, or object-level annotations, which impose substantial burdens on data acquisition, preprocessing, and computation, thereby limiting scalability and transferability. In addition, the recent approach VGLLM (Zheng et al., 2025a) explores a video-based 3D-MLLM setting by adopting the geometry encoder of VGGT (Wang et al., 2025a) to extract 3D geometry features from video.

**3D Reconstruction.** 3D reconstruction has evolved from multi-view geometry pipelines to neural implicit representations and, more recently, feed-forward transformer-based architectures. Classical methods yield accurate geometry but require dense views and heavy preprocessing (Schonberger & Frahm, 2016; Furukawa et al., 2015). Neural radiance fields and point-based extensions improve fidelity and efficiency but focus mainly on appearance modeling while lacking semantic reasoning (Mildenhall et al., 2021; Barron et al., 2022; Kerbl et al., 2023). Recent feed-forward approaches enable direct prediction of depth, pose, and point clouds from video inputs (Wang et al., 2024; Leroy et al., 2024; Wang et al., 2025a). Nevertheless, 3D reconstruction is still only weakly integrated into vision–language research, and its role in supporting semantic reasoning remains underexplored.

**Geometry Priors in Vision-Language Models.** Incorporating geometry priors has become a key approach for Vision-Language Models (VLMs) to enhance spatial understanding. Along a spectrum of reliance on explicit 3D inputs, existing methods can be organized into three categories: first, explicit input injection, which introduces depth, point clouds, or scene graphs as additional modalities to provide metric properties (Cai et al., 2025; Cheng et al., 2024b; Guo et al., 2023); second, internalization at the data and training level, which leverages spatially annotated corpora or geometric distillation to embed geometry implicitly into the alignment space, enabling spatial reasoning without explicit 3D inputs at inference (Chen et al., 2024a; Peng et al., 2023); and third, modular or prompt-based integration, which augments VLMs with lightweight modules or outputs from 3D foundation models, typically without large-scale retraining (Ma et al., 2024; Kerr et al., 2023). In contrast, our approach generates geometry through a video-driven reconstruction branch and achieves alignment with the semantic branch, enabling a structured and reusable integration of geometry priors within a video-based setting.

## 3 METHOD

We present Vid-LLM, a video-based 3D multimodal large language model (3D-MLLM). The main components are presented in the following sections: the Cross-Task Adapter is described in Section 3.1, the reconstruction and reasoning branches are detailed in Sections 3.2 and 3.3, and the training strategy is outlined in Section 3.4. The overall architecture is shown in Fig. 2.

### 3.1 CROSS-TASK ADAPTER

In Vid-LLM, we employ DINOv2 as the shared visual encoder to extract base tokens $T_{base} \in \mathbb{R}^{N \times C}$ from the input image sequence, where $N$ denotes the number of tokens and $C$ is the embedding dimension. To enhance feature effectiveness, we introduce a Cross-Task Adapter (CTA) that aligns 3D geometry priors with vision–language (VL) representations, facilitating the integration of geometric cues into multimodal reasoning.

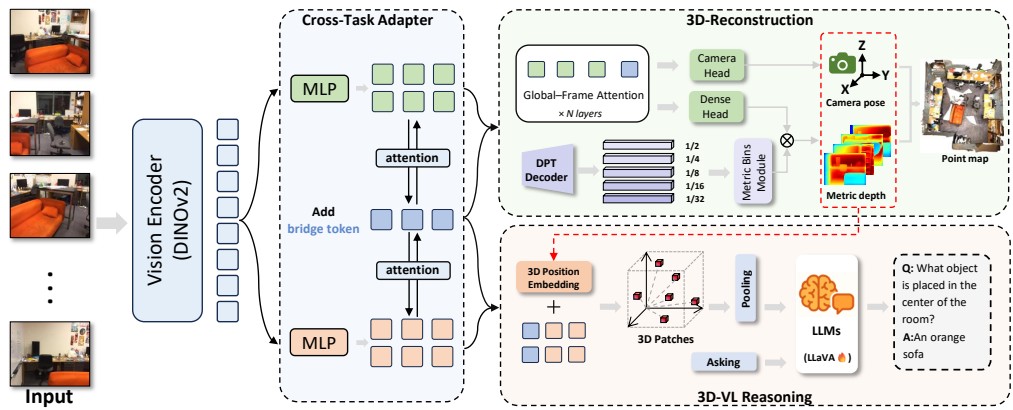

Figure 2: **Architecture of Vid-LLM.** From video, a shared DINOv2 encoder produces tokens that are bidirectionally fused by Cross-Task Adapter with learnable Bridge Tokens, yielding geometric and semantic streams. The reconstruction branch predicts camera poses, depth and recovers real-scale via a Metric-Bins module, while the 3D-VL branch lifts features into 3D tokens for LLM reasoning.

To adapt the shared visual representations for different branches, we employ two lightweight MLP projection heads, $\phi_{geom}(\cdot)$ and $\phi_{lang}(\cdot)$, which map the shared vision tokens to geometry-specific and semantic feature spaces, respectively:

$$T_{geom} = \phi_{geom}(T_{base}), \qquad T_{lang} = \phi_{lang}(T_{base}) \tag{1}$$

To effectively align 3D geometry priors with vision–language representations thus enhance the integration of spatial cues into multimodal reasoning, we introduce learnable Bridge Tokens, denoted as $T_{bridge} \in \mathbb{R}^{K \times C}$. Acting as shared memory units, the bridge tokens attend to geometric and semantic features separately, and the updated representation is formulated as:

$$T'_{bridge} = \text{Attn}(T_{bridge}, T_{geom}^{fused}, T_{geom}^{fused}) + \text{Attn}(T_{bridge}, T_{lang}^{fused}, T_{lang}^{fused}) \tag{2}$$

where $\text{Attn}(\cdot)$ denotes a standard multi-head attention operation. This operation enables bridge tokens to dynamically capture complementary information from both tasks and update their representations during training. The joint propagation of geometric and semantic signals strengthens the alignment of 3D geometry priors with vision–language features, leading to more robust cross-modal representations.

Finally, the updated bridge tokens $T'_{bridge}$ are integrated into the feature streams, yielding enhanced task-specific representations $T'_{geom}$ and $T'_{lang}$. These enriched features capture complementary geometric and semantic cues and are subsequently passed to the reconstruction and reasoning branches. In essence, the Cross-Task Adapter establishes intrinsic geometry–semantics interaction at the feature level, allowing the two streams to reinforce and guide each other for more robust representations.

## 3.2 3D RECONSTRUCTION MODEL

In the reconstruction branch of Vid-LLM, we build on recent transformer-based architectures for end-to-end 3D reconstruction (Wang et al., 2025a) to recover scene geometry from video inputs. To additionally recover real-scale information, we design a Metric Depth Model that provides robust global scale cues, enabling reconstructions with both fine structural details and metric consistency.

**Geometry Encoding and Prediction Heads.** Based on the cross-task enhanced geometric features $T'_{geom}$, together with camera tokens $T^{cam}$ and register tokens $T^{reg}$, the Global-Frame Attention backbone produces an integrated geometric representation, which is then fed into two prediction heads: a camera head estimating intrinsic-extrinsic parameters and a DPT head that predicts the relative depth map $\hat{D}_{rel} \in \mathbb{R}^{H \times W}$, where $H$ and $W$ denote the image height and width, respectively.

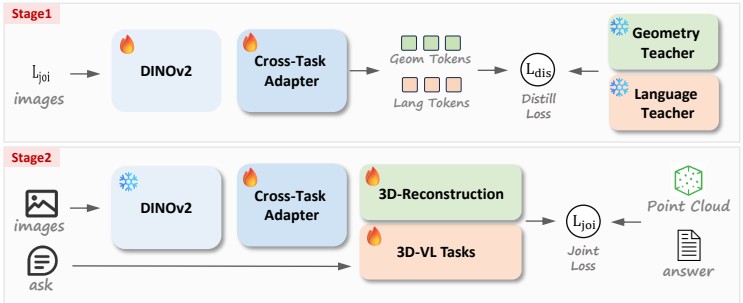

Figure 3: **Overview of the two-stage training strategy.** Stage-1 employs dual-teacher distillation to align geometry and semantics, and Stage-2 jointly optimizes reconstruction and 3D vision–language tasks.

**Metric Depth Model.** To recover real-scale geometry, we equip the DINOv2 features with a DPT-style decoder that produces multi-scale depth representations. Each pixel's depth is modeled using a bin-based formulation, where the probability $p_i(k)$ over the $k-\mathrm{th}$ bin and its refined center $c_i(k)$ jointly determine the prediction as $\hat{d}(i) = \sum_{k=1}^{N} p_i(k)\,c_i(k)$. we use an ordinal-aware normalization to capture relative depth ordering. To further stabilize scale, the bin centers $c_i(k)$ are adaptively refined as $c_i(k) = c_k + \Delta c_i(k)$, where $\Delta c_i(k) = r_k(F_i)$ is predicted from the decoder features $F_i$. The resulting metric depth map $\hat{D}_{\mathrm{metric}} = \{\hat{d}(i)\}_{i=1}^{H \times W}$ provides global scale cues that are aligned with relative predictions for real-scale reconstruction.

**Real-Scale Alignment.** We estimate a scaling factor between the relative depth $\hat{D}_{\mathrm{rel}}$ predicted by the DPT head and the metric depth $\hat{D}_{\mathrm{metric}}$ predicted by the Metric Depth Model via weighted least squares. For each scene, 16 images are randomly sampled to compute per-image scaling factors, and their median is taken median as the final scene-level factor. This factor is then applied to convert both the relative depth and the predicted camera pose into real-world units. Rather than directly using metric depth as the final output, we adopt this alignment strategy since the DPT head provides more accurate texture details compared with the Metric Depth Model, which could also be observed from experimental results.

### 3.3 3D VISION–LANGUAGE MODEL

In the reasoning branch of Vid-LLM, cross-task-enhanced semantic features $T'_{\mathrm{lang}}$ are combined with reconstructed geometry to generate dense 3D patch representations, which are then fed into the LLM for 3D question answering, grounding, and captioning tasks.

**3D Patch Construction.** Each 2D feature $T'_{\mathrm{lang}}(i, j)$ is back-projected into 3D using the estimated depth $\hat{D}$, camera pose $(\hat{R}, \hat{t})$ and intrinsics $K$ produced by the 3D Reconstruction Model, yielding its camera-frame coordinates $P_v(i, j)$ as:

$$P_v(i,j) = \hat{R}^{-1}K^{-1}[i,j,1]^{\top}\hat{D}(i,j) - \hat{R}^{-1}\hat{t} \tag{3}$$

These coordinates are encoded by an MLP into positional embeddings $P'_v(i, j)$ that match the dimensionality of the semantic features. Therefore, the final 3D patch tokens are then obtained by fusing geometry and semantic features:

$$T_{3D}(i,j) = T'_{lang}(i,j) + P'_v(i,j) \tag{4}$$

This operation feeds spatial information into the semantic tokens, further enhancing the spatial awareness in the LLM.

### 3.4 TRAINING STRATEGY

We adopt a two-stage training strategy to utilize the shared encoder for both geometry and semantics. Stage 1 performs dual-teacher distillation, transferring geometric priors from a reconstruction model and semantic knowledge from a multimodal LLM, enabling the encoder to learn both capabilities in a balanced way. Stage 2 jointly optimizes all downstream modules with 3D vision–language

objectives, while incorporating auxiliary reconstruction losses to provide the model with sufficient reconstruction capability and ensure real-scale consistency. The overall pipeline is illustrated in Fig. 3.

**Stage-1 Dual-Teacher Distillation.** In stage 1, we adopt a dual-teacher distillation strategy to jointly train the DINO encoder and the Cross-Task Adapter, enabling the modules to quickly learn both geometric and semantic representations. The pretrained DINO encoder and CLIP encoder serve as the geometry and semantic teachers in the distillation strategy, which are initialized from VGGT (Wang et al., 2025a) and LLaVA-3D (Zhu et al., 2025), respectively. The distillation loss is defined as:

$$L_{distill} = L_{geo}^{feat} + L_{lang}^{feat} + \lambda L_{sc} \tag{5}$$

where $L_{geo}^{feat}$, $L_{lang}^{feat}$ and $L_{sc}$ are the geometry loss, semantic loss and structural consistency loss, respectively. $\lambda$ is a balancing hyperparameter. For geometry learning, an L2 loss is applied to patch-level features, following DINO's patch regression strategy. For semantic learning, a cosine similarity loss is applied to pooled features to align with CLIP's global semantic embedding space. The specific forms are defined as follows:

$$L_{geo}^{feat} = \frac{1}{N} \sum_{i=1}^{N} \|\mathrm{Norm}(T'_{\mathrm{geom},i}) - \mathrm{Norm}(T_{tea,i}^{geo})\|_2^2, \quad L_{lang}^{feat} = 1 - \cos(\mathrm{Pool}(T'_{\mathrm{lang}}), \mathrm{Pool}(T_{tea}^{lang})) \tag{6}$$

where $\mathrm{Norm}(\cdot)$ denotes L2 normalization, and $\mathrm{Pool}(\cdot)$ indicates mean pooling. $T'_{geom}$ and $T'_{lang}$ are the task-specific geometric and semantic features extracted by the Cross-Task Adapter, and serve as the student representations in the distillation strategy. $T_{tea}^{geom}$ and $T_{tea}^{lang}$ are features from the DINO and CLIP encoders and serve as teacher representations. $N$ represents the number of tokens in $T'_{geom}$. Before computing $L_{geo}^{feat}$ and $L_{lang}^{feat}$, we apply lightweight linear projection layers to align the features to the same embedding dimension. To maintain structural consistency, we also introduce the structural consistency loss $L_{sc}$, which is defined as:

$$L_{sc} = \frac{1}{M^2} \|S_{\mathrm{stu}} - S_{\mathrm{tea}}\|_F^2, \quad \text{where } S_{\mathrm{stu}} = Z_{\mathrm{stu}} Z_{\mathrm{stu}}^\top, S_{\mathrm{tea}} = Z_{\mathrm{tea}} Z_{\mathrm{tea}}^\top \tag{7}$$

$Z_{stu} = [\mathrm{Norm}(T'_{\mathrm{geom}}); \mathrm{Norm}(T'_{\mathrm{lang}})] \in \mathbb{R}^{M \times C}$ is obtained by concatenating the geometry and semantic tokens from the student representation. $Z_{tea} = [\mathrm{Norm}(T_{\mathrm{tea}}^{\mathrm{geo}}); \mathrm{Norm}(T_{\mathrm{tea}}^{\mathrm{lang}})]$ is defined in the same way as $Z_{stu}$, but using the teacher representation. $\|\cdot\|_F$ is the Frobenius norm, $M$ is the total number of tokens, and $C$ is the embedding dimension of each token.

**Stage-2 Joint Optimization.** In Stage 2, we further fine-tune all the modules to optimize overall performance. The joint loss is defined as:

$$L_{joint} = L_{recon-task} + L_{VL-task} + L_{MD} \tag{8}$$

where $L_{recon-task}$ is the multi-task loss for 3D reconstruction, consisting of the camera loss, depth loss, and point map loss following Wang et al. (2025a). $L_{VL-task}$ supervises 3D vision–language reasoning, including cross-entropy loss for instruction-following tasks, along with bounding box regression and matching losses for grounding (Zhu et al. (2025)). $L_{MD}$ represents the metric depth loss, combining a global scale penalty and a robust local refinement term, and is defined as:

$$L_{\mathrm{MD}} = b^2 + \frac{1}{K} \sum_{i=1}^{K} \frac{(e_i - b)^2}{1 + \alpha|e_i - b|}, \tag{9}$$

The log-depth error is defined as $e_i = \log(d_i^{pred} + \varepsilon) - \log(d_i^{gt} + \varepsilon)$, where $\varepsilon$ is a small constant for numerical stability. $b = \frac{1}{K} \sum_{i=1}^{K} e_i$ is the mean error across all $K$ valid pixels in the image. The parameter $\alpha > 0$ controls the robustness by down-weighting large residuals.

During joint training, $L_{recon-task}$ and $L_{MD}$ optimize the 3D reconstruction model, $L_{VL-task}$ optimize the 3D vision-language model, while the shared CTA is jointly optimized by $L_{recon-task}$ and $L_{VL-task}$. It is worth to noticing that the loss constructed in the 3D-VL reasoning branch is not used to optimize the 3D reconstruction model, which can be seen in Fig. 2. This one-way gradient flow ensures that the CTA acts as a stable bridge for geometry-semantics interaction, enabling effective feature exchange.

Table 1: **Evaluation of 3D Question Answering on ScanQA and SQA3D.** Methods marked with * are 3D MLLM evaluated in video mode. [†] indicates the model consumes VGGT-generated 3D geometry. "C" stands for "CIDEr", "B-4" for "BLEU-4", "M" for "METEOR", "R" for "ROUGE", and "EM@1" for top-1 exact match.

| Method | ScanQA | | | | | SQA3D |
|---|---|---|---|---|---|---|
| | C↑ | B-4↑ | M↑ | R↑ | EM@1↑ | EM@1↑ |
| *3D-based models* | | | | | | |
| Scan2Cap | – | – | – | – | – | 41.0 |
| ScanQA | 64.9 | 10.1 | 13.1 | 33.3 | 21.1 | 47.2 |
| 3D-VisTA | 69.6 | 10.4 | 13.9 | 35.7 | 22.4 | 48.5 |
| 3D-LLM | 69.4 | 12.0 | 14.5 | 35.7 | 20.5 | – |
| LL3DA | 76.8 | 13.5 | 15.9 | 37.3 | – | – |
| Grounded3D-LLM | 75.9 | 13.2 | – | – | – | – |
| Chat-3D v2 | 77.1 | 7.3 | 16.1 | 40.1 | 21.1 | – |
| Scene-LLM | 80.0 | 12.0 | 16.6 | 40.0 | 27.2 | 54.2 |
| ChatScene | 87.7 | 14.3 | 18.0 | 41.6 | 21.6 | 54.6 |
| LEO | 101.4 | 13.2 | 20.0 | 49.2 | 24.5 | 50.0 |
| Video-3D LLM | 102.6 | 16.2 | 19.8 | 49.0 | 30.1 | 58.6 |
| LLaVA-3D | 103.1 | 16.4 | **20.8** | 49.6 | **30.6** | 60.1 |
| 3DRS | **104.8** | **17.2** | 20.5 | **49.8** | 30.3 | **60.6** |
| *Video-based models* | | | | | | |
| VILA-40B | 48.2 | 9.9 | 11.4 | 27.3 | 17.2 | 37.2 |
| IXC-2.5-7B | 53.9 | 8.7 | 13.2 | 31.5 | 19.4 | 41.5 |
| LLaVA-OV-7B | 78.2 | 13.3 | 17.2 | 40.5 | 22.9 | 47.6 |
| LLaVA-Video-7B | 88.7 | – | – | – | – | 48.5 |
| Uni3DR[2]* | 70.3 | 12.2 | 14.9 | 36.3 | 17.3 | – |
| ChatScene* | 85.6 | – | – | – | – | 52.9 |
| 3DRS[†] | 94.7 | 12.3 | 15.9 | 45.1 | 23.9 | 54.5 |
| Vid-LLM (Ours) | **101.9** | **15.8** | **18.3** | **49.5** | **27.6** | **57.3** |

Table 2: **Evaluation of 3D Dense Captioning on Scan2Cap.** The n-gram metrics for Scan2Cap are governed by IoU@0.5.

| Method | Scan2Cap | | | |
|---|---|---|---|---|
| | C@0.5↑ | B-4@0.5↑ | M@0.5↑ | R@0.5↑ |
| *3D-based models* | | | | |
| Scan2Cap | 39.1 | 23.3 | 22.0 | 44.8 |
| Grounded3D-LLM | 70.2 | 35.0 | – | – |
| LEO | 68.4 | 36.9 | 27.7 | 57.8 |
| ChatScene | 77.1 | 36.3 | 28.0 | 58.1 |
| Video-3D LLM | 83.8 | 42.4 | 28.9 | 62.3 |
| LLaVA-3D | 84.1 | **42.6** | **29.0** | **63.4** |
| 3DRS | **86.1** | 41.6 | 28.9 | 62.3 |
| *Video-based models* | | | | |
| 3DRS[†] | 76.4 | 39.6 | 27.3 | 57.1 |
| VGLLM | 78.6 | **40.9** | 28.6 | **62.4** |
| Vid-LLM (Ours) | **81.5** | **40.9** | **28.7** | 61.8 |

Table 3: **Evaluation of 3D Visual Grounding on ScanRefer and Multi3DRefer.**

| Method | ScanRefer | | Multi3DRefer | |
|---|---|---|---|---|
| | Acc@0.25↑ | Acc@0.5↑ | Acc@0.25↑ | Acc@0.5↑ |
| *3D-based models* | | | | |
| ScanRefer | 37.3 | 24.3 | – | – |
| Chat-3D v2 | 35.9 | 30.4 | – | – |
| Grounded3D-LLM | 48.6 | 44.0 | 44.7 | 40.8 |
| LLaVA-3D | 50.1 | 42.7 | 49.8 | 43.6 |
| ChatScene | 55.5 | 50.2 | 57.1 | 52.4 |
| Video-3D LLM | 58.1 | 51.7 | 58.0 | 52.7 |
| 3DRS | **62.9** | **56.1** | **60.4** | **54.9** |
| *Video-based models* | | | | |
| VGLLM | 53.5 | 47.5 | – | – |
| 3DRS[†] | 55.2 | 51.1 | 52.8 | 48.3 |
| Vid-LLM (Ours) | **63.2** | **56.4** | **61.6** | **56.1** |

## 4 EXPERIMENTS

This section provides a comprehensive evaluation of Vid-LLM. In Section 4.1, we introduce the experimental setup, covering the training details of Vid-LLM and the evaluation settings used in our experiments. In Section 4.2, we evaluate the model on 3D vision–language reasoning tasks against state-of-the-art methods. In Section 4.3, we compare Vid-LLM with representative joint reconstruction–reasoning models to examine how different design choices affect performance when jointly executing reconstruction and 3D VL reasoning. In Section 4.4, we conduct ablation studies to analyze the contributions of core modules.

### 4.1 EXPERIMENTAL SETUP

**Training Details.** We adopt DINOv2-L as the visual backbone, consisting of 24 Transformer layers with a hidden dimension of 1024. The projection MLPs use two fully connected layers with an expansion factor of 4 and GELU activation. For data processing, we uniformly sample 32 frames per scene, resize the shorter side to 518, crop the resolution so that both height and width are multiples of 14. Optimization is performed using AdamW with $\beta_1 = 0.9$, $\beta_2 = 0.999$, and weight decay of 0.05, with gradient clipping at 1.0.

**Evaluation Details.** We follow the standard evaluation protocols and metrics defined for each dataset. For tables that summarize performance across multiple datasets (e.g., Tables 5 and 6), the reported numbers are computed as the mean of the metrics for each benchmark. For the Scan2Cap dataset, we follow prior work (Zhu et al., 2025; Zheng et al., 2025a): instance proposals are generated using Mask3D (Schult et al., 2022), and 3D coordinate tokens are constructed from the predicted instance centers to enable instance-aware caption generation. For the ScanRefer dataset, Mask3D (Schult et al., 2022) serves as the 3D segmentor to remain consistent with supervised baselines (Huang et al., 2024; Chen et al., 2024c; Huang et al., 2023a). For the Nr3D and Sr3D datasets, the provided segmentation annotations are adopted to align with previous methods (Chen et al., 2022; Huang et al., 2023a). For reconstruction evaluation on ScanNet, the metrics are computed in metric scale. All models are trained with the same data and settings for a fair comparison.

## 4.2 3D VISION–LANGUAGE REASONING

**Overview.** To comprehensively assess the performance of Vid-LLM on 3D vision–language reasoning tasks, we conduct experiments using widely adopted datasets covering three task categories: 3D Question Answering on ScanQA (Azuma et al., 2022) and SQA3D (Ma et al., 2022), 3D Dense Captioning on Scan2Cap (Chen et al., 2021), and 3D Visual Grounding on ScanRefer (Chen et al., 2020), Multi3DRefer (Zhang et al., 2023), and Nr3D/Sr3D (Achlioptas et al., 2020b).

**Baseline.** We compare Vid-LLM with a broad set of 3D-based and video-based vision–language reasoning models. 3D-based methods rely on explicit 3D scene inputs such as point clouds or reconstructed geometry, whereas video-based methods operate solely on video. The 3D-based baselines include ScanQA (Azuma et al., 2022), Scan2Cap (Chen et al., 2021), 3D-VisTA (Zhu et al., 2023), 3D-LLM (Hong et al., 2023b), LL3DA (Chen et al., 2024b), Grounded3D-LLM (Chen et al., 2024c), Chat-3D v2 (Huang et al., 2023a), LEO (Huang et al., 2023b), Scene-LLM (Fu et al., 2024), ChatScene (Huang et al., 2024), LLaVA-3D (Zhu et al., 2025), Video-3D LLM (Zheng et al., 2025b), and 3DRS (Huang et al., 2025). The video-based baselines include VILA-40B (Lin et al., 2024), IXC (Zhang et al., 2024b), LLaVA-OV (Li et al., 2024), LLaVA-Video (LLaVA Team, 2024), Uni3DR2 (Chu et al., 2024), and VGLLM (Zheng et al., 2025a). To further assess the grounding capability of Vid-LLM, we additionally compare against task-specific 3D visual grounding models, including ScanRefer (Chen et al., 2020), 3D-VisTA (Zhu et al., 2023), ReferIt3D (Achlioptas et al., 2020a), 3DVG-Trans (Zhao et al., 2021), MVT (Huang et al., 2022), ViL3DRel (Chen et al., 2022), and SceneVerse (Jia et al., 2024). To allow comparison under the same video input setting with the most competitive 3D-based baseline in our evaluation, we additionally include 3DRS[†], a variant of 3DRS that takes VGGT-generated 3D geometry as input. These comparisons collectively provide a comprehensive evaluation of Vid-LLM across 3D vision–language reasoning tasks.

**Result & Analysis.** Vid-LLM consistently shows robust performance across all 3D vision–language reasoning benchmarks. As shown in Tab. 1, Vid-LLM achieves the best results among video-based models on both ScanQA and SQA3D, outperforming the second-best baseline (3DRS[†]) by an average margin of 11%. Tab. 2 further shows that Vid-LLM attains the highest performance among video-based methods on Scan2Cap; notably, its M@0.5 score (28.7) is close to that of the best-performing 3D-based model (29.0), despite not using depth or point clouds. These results collectively indicate that the integration of geometric cues into semantic reasoning can significantly improve perception performance. Tab. 3 and Tab. 4 present results on ScanRefer/Multi3DRefer and Nr3D/Sr3D. Vid-LLM achieves the best performance across all metrics on the two grounding benchmarks, surpassing all comparable 3D-based and video-based counterparts. These results demonstrate that the proposed Cross-Task Adapter, which effectively integrates semantic and geometric information, enables effective spatial reasoning and yields substantial gains on geometry-intensive tasks such as 3D visual grounding. For qualitative analysis, we visualize the 3D grounding results on the ScanRefer dataset in Fig. 4, which further demonstrates the 3D visual grounding capability of Vid-LLM. More qualitative visualizations for 3D VL tasks can be found in Appendix A.1.

## 4.3 COMPARISON WITH JOINT RECONSTRUCTION–REASONING MODELS

**Overview.** In addition to 3D vision–language reasoning, Vid-LLM also incorporates a reconstruction branch that provides geometric priors and metric-scale cues. This naturally raises the question of how different design choices for coupling reconstruction and reasoning affect performance when both tasks must be performed from a single video input. To provide a comprehensive analysis, we compare Vid-LLM with two representative categories of joint reconstruction–reasoning baselines: (i) simple concatenation pipelines that feed the 3D geometry reconstructed from video into a 3D multimodal LLM, and (ii) end-to-end architectures that integrate reconstruction and 3D VL reasoning within a single model.

**Baseline.** For the concatenation baselines, we construct pipelines that feed the predicted geometry information of VGGT into LLaVA-3D (Zhu et al., 2025) for 3D vision–language reasoning. Three variants of the concatenation pipeline are evaluated: (i) VGGT+LLaVA-3D, the direct combination of VGGT and LLaVA-3D; (ii) VGGT+LLaVA-3D[†], which introduces metric-scale alignment to the predictions of VGGT; (iii) VGGT+LLaVA-3D[†‡], which further fine-tunes LLaVA-3D using depth and camera poses predicted by VGGT as geometric supervision. For end-to-end joint-task baselines, we include Uni3DR[2] (Chu et al., 2024), an end-to-end framework that integrates a reconstruction

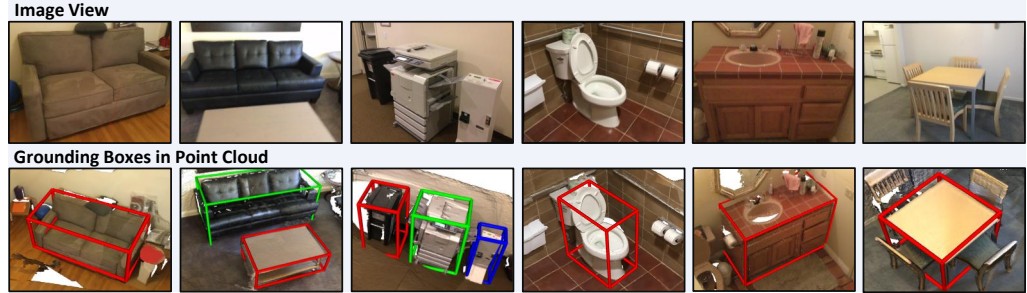

Figure 4: **Qualitative results of 3D visual grounding on the ScanRefer dataset.** The predicted 3D bounding boxes are visualized on point clouds reconstructed by our model.

Table 4: **Evaluation of 3D Visual Grounding on Nr3D and Sr3D.** Results are reported in grounding accuracy (%).

| Method | Nr3D | | | Sr3D | | |
|---|---|---|---|---|---|---|
| | Overall↑ | Hard↑ | View Dep↑ | Overall↑ | Hard↑ | View Dep↑ |
| ScanRefer | 34.2 | 23.5 | 29.9 | – | – | – |
| ReferIt3D | 35.6 | 27.9 | 32.5 | 40.8 | 31.5 | 39.2 |
| 3DVG-Trans | 40.8 | 34.8 | 34.8 | 51.4 | 44.9 | 44.6 |
| MVT | 55.1 | 49.1 | 54.3 | 64.5 | 58.8 | 58.4 |
| ViL3DRel | 62.6 | 55.6 | 59.8 | 72.5 | 68.1 | 58.0 |
| Chat-3D v2 | 63.6 | 57.0 | 61.1 | 73.1 | 68.4 | 58.1 |
| 3D-VisTA | 64.2 | 56.7 | 61.5 | 76.4 | 71.3 | 58.9 |
| SceneVerse | 64.9 | 57.8 | 56.9 | 77.5 | 71.6 | 62.8 |
| Vid-LLM (Ours) | **65.4** | **57.9** | **61.9** | **77.8** | **72.2** | **63.1** |

Table 5: **Comparison with end-to-end frameworks for joint 3D reconstruction and 3D VL reasoning.**

| Method | Recon | 3D VL Tasks | | | Time |
|---|---|---|---|---|---|
| | ScanNet↑ | ScanQA↑ | Scan2Cap↑ | ScanRefer↑ | (s / scene)↓ |
| *Concatenation models* | | | | | |
| VGGT+LLaVA-3D | – | 24.6 | 35.1 | 31.6 | 2.7 |
| VGGT+LLaVA-3D[†] | **0.591** | 42.1 | 50.9 | 47.3 | 2.7 |
| VGGT+LLaVA-3D[†‡] | **0.591** | 41.6 | 48.2 | 45.9 | 2.7 |
| *Joint-task models* | | | | | |
| Uni3DR[2] | 0.580 | 30.2 | 42.7 | 33.6 | 2.1 |
| VGLLM-Rec | 0.541 | 40.9 | 51.6 | 48.5 | 1.8 |
| Vid-LLM (Ours) | 0.582 | **45.7** | **53.2** | **59.8** | **1.6** |

† denotes introducing real-scale;

‡ denotes training LLaVA-3D with 3D geometry predicted by VGGT

module with a 3D VL reasoning branch. Notably, Uni3DR$^2$ requires ground-truth camera poses as input, whereas Vid-LLM performs both tasks directly from video, making the comparison conservative in favor of Uni3DR$^2$. We also include VGLLM-Rec, an extension of VGLLM (Zheng et al., 2025a). To support joint reconstruction and reasoning, we reconnect the original camera and depth prediction heads of VGGT to the geometry encoder in VGLLM, enabling VGLLM-Rec to generate geometric predictions in addition to performing 3D VL reasoning from video.

**Result & Analysis.** As shown in Tab. 5, VGGT+LLaVA-3D obtains very low 3D VL accuracy, indicating that relative-scale geometry cannot provide reliable cues for 3D VL reasoning. Building on this, VGGT+LLaVA-3D[†] shows a clear performance gain once metric-scale alignment is applied. However, even with this setting, the concatenation baseline still underperforms Vid-LLM on all 3D VL tasks despite achieving a higher reconstruction score (0.591 vs. 0.582). This indicates that simply providing accurate reconstructed geometry to a 3D-LLM is insufficient and that effective geometry–semantics interaction is necessary for reliable reasoning performance. Furthermore, VGGT+LLaVA-3D[†‡] yields a slight drop in 3D VL accuracy compared with VGGT+LLaVA-3D[†], suggesting that training the 3D-LLM with noisy predicted geometry may introduce biases that degrade the semantic representations of the framework. In contrast to concatenation pipelines, end-to-end joint-task models avoid multi-stage processing and achieve lower inference latency, reducing runtime from 2.7 s/scene to 1.6–2.1 s/scene. From Tab. 5, we can observe that Vid-LLM achieves the best overall performance among the joint models on both reconstruction and 3D VL tasks. It marginally outperforms Uni3DR² in reconstruction quality on ScanNet (0.582 vs. 0.580) despite Uni3DR² having access to ground-truth camera poses. On the 3D VL tasks, Vid-LLM demonstrates a clear advantage, with an average relative improvement of 47.6% over Uni3DR² and 12.1% over VGLLM-Rec across ScanQA, Scan2Cap, and ScanRefer. Overall, Vid-LLM delivers the fastest inference speed and the best 3D VL performance, while also achieving the best reconstruction quality among joint-task models. These gains stem from our integrated architecture and the Cross-Task Adapter, which facilitates geometry–semantics interaction and allows the model to effectively leverage geometric cues during joint reasoning. More experimental results about reconstruction performance can be found in Appendix A.2.

Table 6: **Ablation on Cross-Task Adapter (CTA) and Metric Depth (MD) Modules.**

| | 3D VL Tasks | | | 3D Recon |
|---|---|---|---|---|
| | ScanOA ↑ | Scan2Cap ↑ | ScanRefer ↑ | scannet ↑ |
| w/o CTA | 33.7 | 36.6 | 34.1 | 0.402 |
| CTA – Concat-SA | 38.2 | 41.9 | 39.6 | 0.461 |
| CTA – w/o Bridge (CA) | 40.8 | 45.6 | 42.2 | 0.473 |
| CTA – 4tokens | 43.8 | 50.9 | 47.8 | 0.501 |
| CTA – 8tokens | 44.1 | 52.7 | 48.1 | 0.529 |
| CTA – 16tokens (Ours) | **45.7** | **53.2** | **48.4** | **0.582** |
| CTA – 32tokens | 45.2 | 52.8 | 47.6 | 0.564 |
| w/o MD | 29.6 | 35.7 | 33.9 | - |
| MD – w/o Alignment | 42.1 | 49.6 | 43.2 | 0.531 |
| MD (Ours) | **45.7** | **53.2** | **48.4** | **0.582** |

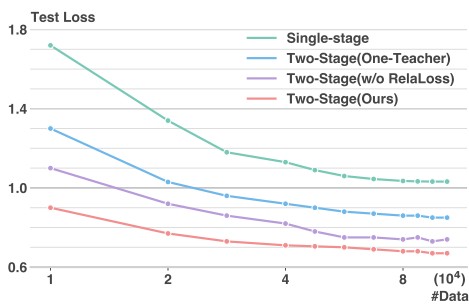

Figure 5: Test loss vs. data size across training strategies.

## 4.4 ABLATION STUDIES

**Cross-Task Adapter.** To assess the contribution of the Cross-Task Adapter, we compare several configurations: removing the CTA module (w/o CTA); in the setting without bridge tokens, applying self-attention to the concatenated $T_{geom}$ and $T_{lang}$ (CTA–Concat-SA) and performing cross-attention between the two feature sets (CTA–w/o Bridge (CA)); and finally using the full CTA with bridge tokens. We also analyze how different numbers of bridge tokens (4, 8, 16, and 32) affect model performance. As shown in Tab. 6, w/o CTA leads to the lowest performance on both tasks, confirming that a shared visual representation alone cannot jointly support reconstruction and 3D VL reasoning. Both CTA–Concat-SA and CTA–w/o Bridge(CA) show clear improvements, yet their effectiveness is limited as neither design provides a stable shared latent space for passing geometric cues to semantic features. The configurations using the full CTA module achieve clear performance improvements, since the bridge tokens provide a dedicated latent space that enables consistent geometry–semantic alignment and more effective cross-task interaction. Among different token counts, using 16 bridge tokens achieves the best balance between accuracy and model complexity.

**Metric Depth Modules.** Tab. 6 also reports ablations on the use of metric depth and our alignment strategy. Without metric depth supervision (w/o MD), scale ambiguity destroys geometric consistency and leads to near failure of 3D VL reasoning. Incorporating metric depth without alignment (MD – w/o Alignment) preserves metric scale but produces less accurate reconstruction, which in turn limits the performance of 3D VL tasks. By contrast, combining metric depth with our scale alignment strategy effectively exploits both global scale cues and relative structural details, yielding the most accurate reconstruction and consistently more reliable results on 3D VL tasks.

**Two-Stage Training.** Fig. 5 compares different training strategies to assess their impact on convergence and performance. Training all modules jointly without staging (Single-stage) converges slowly and stabilizes at a relatively high loss, due to gradient interference across modules. Using a single teacher under the two-stage strategy (Two-Stage (One-Teacher)) offers limited supervision, resulting in less effective optimization. Removing the relational loss during two-stage strategy (Two-Stage (w/o RelaLoss)) further slows convergence and reduces accuracy, highlighting the importance of enforcing cross-task consistency. In contrast, using the full two-stage strategy with dual-teacher supervision and relational consistency (Two-Stage (Ours)) yields the best training performance.

## 5 CONCLUSION

In this work, we present Vid-LLM, a video-based 3D Multimodal Large Language Model (3D-MLLM). Our compact architecture extracts geometric cues from video and feeds them into the LLM through a 3D patch construction strategy to accomplish spatial reasoning. A central component of our framework is the Cross-Task Adapter that aligns 3D geometry priors with vision–language representations. This design enhances their integration into the MLLM and improves the robustness of reasoning under uncertain geometry. With a two-stage training strategy, our model achieves greater training stability and faster convergence. Extensive experiments on 3D vision–language benchmarks demonstrate that Vid-LLM achieves state-of-the-art results on several benchmarks and remains competitive on the others, while ablation studies validate the effectiveness of each component.

## 6 REPRODUCIBILITY STATEMENT

We are committed to ensuring reproducibility. Part of our implementation is provided in the Supplementary Materials. Upon acceptance, we will release the complete code, datasets, and pretrained checkpoints to enable full verification of our results.

## 7 ACKNOWLEDGMENTS

This work was supported by the National Natural Science Foundation of China (Grant No. 42374016), the Major Program (JD) of Hubei Province (Grant No. 2023BAA025), and the Open Fund of Hubei Luojia Laboratory (Grant No. 260100023). The numerical calculations in this paper have been done on the supercomputing system in the Supercomputing Center of Wuhan University.

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

# A APPENDIX

In the Appendix, we provide the following experimental results:

- qualitative results and challenging cases of Vid-LLM in Appendix A.1;
- supplementary experiments on reconstruction performance in Appendix A.2;
- the use of large language models (LLMs) statement in Appendix A.3.

## A.1 QUALITATIVE RESULTS AND CHALLENGING CASES

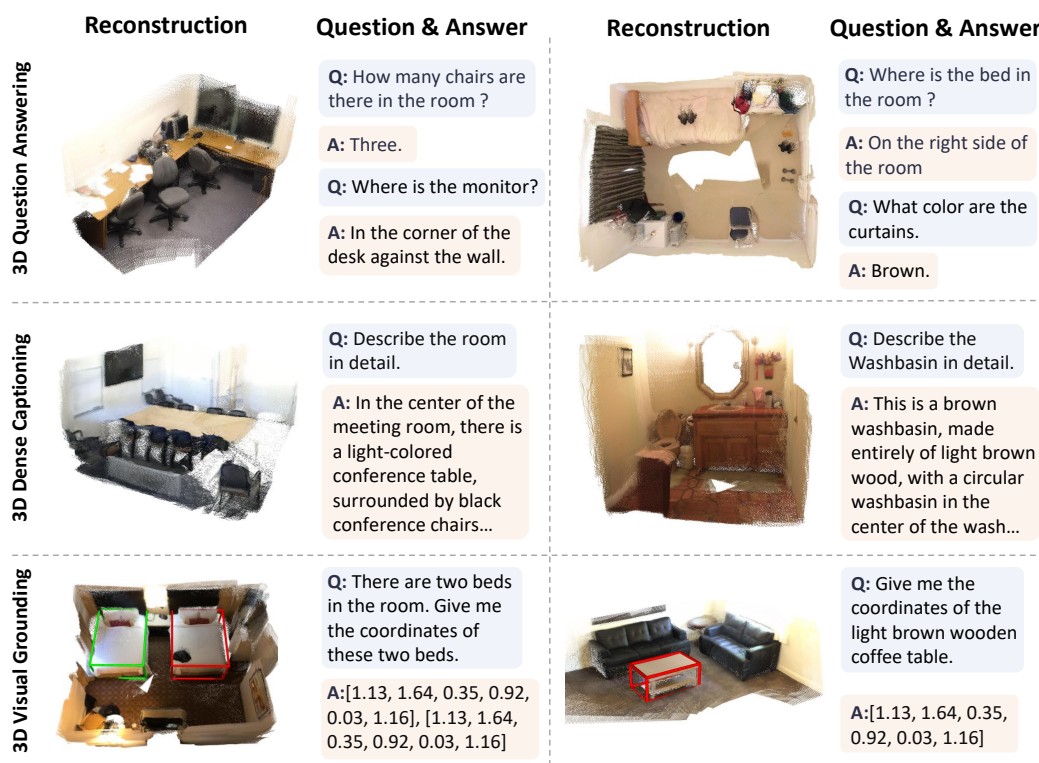

Figure 6: **Qualitative Results.**

**Qualitative Results.** Fig. 6 presents reconstruction results on the ScanNet dataset, together with three representative 3D vision–language (3D VL) tasks. Specifically, it illustrates (1) 3D Question Answering, where the model answers queries on object counts, locations, and attributes from reconstructed scenes; (2) 3D Dense Captioning, which provides detailed semantic descriptions of rooms and key objects; and (3) 3D Visual Grounding, where the model localizes target objects in 3D space according to textual instructions. These qualitative results demonstrate the capability of Vid-LLM in scene reconstruction and 3D VL reasoning from video inputs.

**Challenging Cases.** Fig. 7 presents three challenging cases for Vid-LLM on 3D VL tasks. These scenes reflect typical failure modes of monocular video reconstruction, where limited camera viewpoints, reflective surfaces, or oblique viewing angles lead to incomplete geometry. As shown in examples (a) and (b), tasks that rely on explicit 3D structural information, such as instance counting or estimating 3D object extents for grounding, are directly affected by incomplete or unreliable geometry. In contrast, example (c) shows that Vid-LLM remains effective when sufficient 2D cues are present, due to the Cross-Task Adapter and bridge-token mechanism, which jointly align semantic and geometric information. These observations suggest that improving the reconstruction branch to enhance geometric fidelity could further strengthen the overall 3D VL performance, indicating a promising direction for future work.

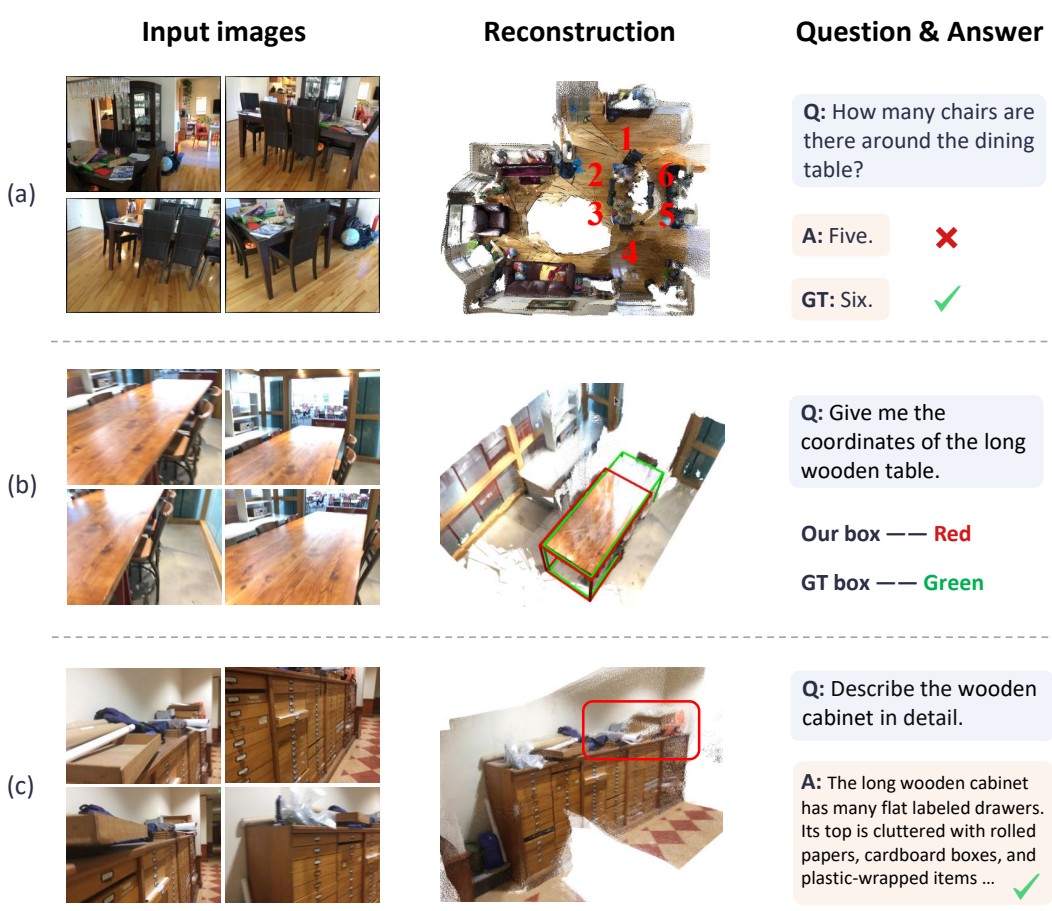

Figure 7: **Challenging Cases.** We present one challenging case for each 3D VL task—3D question answering (a), 3D visual grounding (b), and 3D dense captioning (c). (a) Due to limited camera viewpoints in the input video, one chair is only partially observed and consequently missing from the reconstruction, leading to an undercounted result. (b) The highly reflective tabletop results in unstable depth estimates and incomplete reconstruction, resulting in a predicted 3D box that is smaller than the ground truth. (c) The top surface of the cabinet is not well reconstructed because it appears only briefly and mostly from oblique angles, yet Vid-LLM nonetheless produces an accurate caption by leveraging rich 2D semantic cues.

Table 7: **Camera pose evaluation on Co3Dv2 and RealEstate10K.**

| Method | Co3Dv2 | | | RealEstate10K |
|---|---|---|---|---|
| | RRA@15↑ | RTA@15↑ | mAA(30)↑ | mAA(30)↑ |
| COLMAP+SG (Sarlin et al., 2020) | 36.1 | 27.3 | 25.3 | 45.2 |
| PixSfM (Lindenberger et al., 2021) | 33.7 | 32.9 | 30.1 | 49.4 |
| PoseDiff (Lee & Lee, 2024) | 80.5 | 79.8 | 66.5 | 48.0 |
| DUSt3R (Wang et al., 2024) | 94.3 | 88.4 | 77.2 | 61.2 |
| MASt3R (Leroy et al., 2024) | 94.6 | 91.9 | 81.8 | 76.4 |
| FLARE (Zhang et al., 2025) | 95.4 | 83.6 | 78.4 | 75.3 |
| Fast3R (Yang et al., 2025) | 96.2 | 81.6 | 75.0 | 72.7 |
| CUT3R (Wang et al., 2025b) | 95.7 | 84.5 | 73.3 | 77.1 |
| VGGT (Wang et al., 2025a) | **97.3** | 93.3 | **89.6** | **83.8** |
| Vid-LLM (Ours) | 96.8 | **93.4** | 88.5 | 83.1 |

Table 8: **Depth estimation results on the NYU Depth v2 dataset.** [†] Model outputs are in relative scale; we align them to the ground-truth metric scale prior to evaluation.

| Method | $\delta 1 \uparrow$ | AbsRel $\downarrow$ | RMSE $\downarrow$ | log10 $\downarrow$ |
|---|---|---|---|---|
| DPT (Ranftl et al., 2021) | 0.904 | 0.110 | 0.357 | 0.045 |
| P3Depth (Patil et al., 2022) | 0.898 | 0.104 | 0.356 | 0.043 |
| SwinV2-L (Liu et al., 2022) | 0.949 | 0.083 | 0.287 | 0.035 |
| AiT (Ning et al., 2023) | 0.954 | 0.076 | 0.275 | 0.033 |
| VPD (Zhao et al., 2023) | 0.964 | 0.069 | 0.254 | 0.030 |
| ZoeDepth (Bhat et al., 2023) | 0.953 | 0.075 | 0.270 | 0.032 |
| DepthAnything (Yang et al., 2024) | 0.984 | 0.056 | 0.206 | 0.024 |
| VGGT[†] (Wang et al., 2025a) | 0.989 | 0.022 | 0.103 | 0.011 |
| Vid-LLM(Ours) | **0.987** | **0.025** | **0.109** | **0.010** |

Table 9: **Reconstruction results on the ScanNet dataset.** [†] Model outputs are in relative scale; we align them to the ground-truth metric scale prior to evaluation.

| Method | GT cams | Comp $\downarrow$ | Acc $\downarrow$ | Recall $\uparrow$ | Prec $\uparrow$ | F-score $\uparrow$ |
|---|---|---|---|---|---|---|
| MVDNet (Ou et al., 2022) | $\checkmark$ | 0.040 | 0.240 | 0.831 | 0.208 | 0.329 |
| GPMVS (Liu et al., 2023) | $\checkmark$ | **0.031** | 0.879 | **0.871** | 0.188 | 0.304 |
| Atlas (Murez et al., 2020) | $\checkmark$ | 0.062 | 0.128 | 0.732 | 0.382 | 0.499 |
| NeuralRecon (Sun et al., 2021) | $\checkmark$ | 0.106 | 0.073 | 0.428 | 0.592 | 0.494 |
| DG-Recon (Ju et al., 2023) | $\checkmark$ | 0.085 | **0.039** | 0.476 | 0.675 | 0.521 |
| PanoRecon (Wu et al., 2024) | $\checkmark$ | 0.089 | 0.064 | 0.530 | 0.656 | **0.584** |
| RCVD (Kopf et al., 2021) | $\times$ | 0.161 | 0.425 | 0.164 | 0.109 | 0.125 |
| GCVD (Lee et al., 2022) | $\times$ | 0.175 | 0.278 | 0.178 | 0.146 | 0.147 |
| Colmap (Schonberger & Frahm, 2016) | $\times$ | 0.142 | 0.367 | 0.119 | 0.267 | 0.178 |
| FrozenRecon (Xu et al., 2023) | $\times$ | 0.092 | 0.085 | 0.436 | 0.336 | 0.410 |
| DUSt3R (Wang et al., 2024) | $\times$ | 0.243 | 0.179 | 0.284 | 0.657 | 0.387 |
| VGGT[†] (Wang et al., 2025a) | $\times$ | 0.067 | 0.044 | 0.658 | 0.891 | 0.580 |
| Vid-LLM(Ours) | $\times$ | **0.071** | **0.042** | **0.634** | **0.885** | **0.582** |

## A.2 SUPPLEMENTARY EXPERIMENTS ON RECONSTRUCTION

**Overview.** Since the reconstruction branch not only provides 3D cues for the 3D-MLLM but also explicitly produces point clouds that can serve downstream applications, it is important to assess its individual performance. To evaluate the 3D modeling capability of Vid-LLM, we assess three tasks: camera pose estimation, depth prediction, and point-cloud reconstruction. Depth prediction is crucial to real-scale reconstruction, and together with poses it determines the quality of the recovered point clouds. Following standard protocols, we evaluate poses on Co3Dv2 (Reizenstein et al., 2021) and RealEstate10K (Zhou et al., 2018), depth on the indoor NYU Depth v2 dataset (Silberman et al., 2012), and point-cloud reconstruction on ScanNet (Dai et al., 2017), ensuring consistency with the 3D vision–language benchmarks introduced earlier. Depth and reconstruction evaluations require the model to predict real-scale geometric outputs, while camera pose estimation is scale-invariant.

**Baseline.** For 3D reconstruction, we evaluate three subtasks: (i) camera pose estimation, where we compare against geometry-based optimization methods including COLMAP+SG and PixSfM, as well as state-of-the-art learning-based approaches such as PoseDiff, DUSt3R, MASt3R, FLARE, Fast3R, and CUT3R; (ii) monocular depth estimation, where we benchmark against DPT, P3Depth, SwinV2-L, AiT, VPD, ZoeDepth, and DepthAnything; and (iii) point-cloud reconstruction, where we consider pose-dependent methods such as MVDNet, GPMVS, Atlas, NeuralRecon, DG-Recon, and PanoRecon, together with pose-free approaches including RCVD, GCVD, COLMAP, Frozen-Recon, and DUSt3R, ranging from traditional multi-view geometry to modern learning-based techniques. We further benchmark against VGGT, which serves as both a high-performing reconstruction model and as the reconstruction teacher in our framework.

**Result & Analysis.** Across the three benchmarks, Vid-LLM achieves reconstruction quality comparable to VGGT while operating in an end-to-end real-scale setting. On camera pose estimation (Tab. 7), Vid-LLM attains results close to VGGT on both Co3Dv2 and RealEstate10K and slightly

outperforms it on RTA@15 for Co3Dv2. On NYU Depth v2 and ScanNet (Tab. 8 and Tab. 9), VGGT requires an explicit scale-alignment step to match the metric ground truth, whereas Vid-LLM predicts real-scale depth and point clouds in an end-to-end manner. In this setting, Vid-LLM achieves lower log10 error on NYU and higher F-score and accuracy on ScanNet than VGGT, indicating that its real-scale geometry remains competitive against the teacher model. Vid-LLM also surpasses all other models that predict real-scale outputs. Overall, these results confirm that its reconstruction module delivers consistent and reliable geometric performance.

### A.3 THE USE OF LARGE LANGUAGE MODELS (LLMS) STATEMENT

Parts of the language in this manuscript were polished with the assistance of Large Language Models (LLMs). The authors have carefully reviewed and verified all LLM-assisted text to ensure accuracy and appropriateness. The intellectual contributions, ideas, and conclusions presented in this work are entirely those of the authors.

