# OpenReview forum: "Vid-LLM: A Compact Video-based 3D Multimodal LLM with Reconstruction–Reasoning Synergy"
_ICLR.cc/2026/Conference — ICLR 2026 Oral_

### Official Review · Reviewer_1jxT · 2025-10-31

**Soundness:** 3
**Presentation:** 3
**Contribution:** 3
**Rating:** 6
**Confidence:** 4

**Summary:**

The paper introduces Vid-LLM, a video-based Multimodal Large Language Model for 3D scene understanding that works directly from video, removing the need for external 3D data. The method leverages geometric priors and incorporates them efficiently using a Cross-Task Adapter (CTA) to align 3D geometry with vision-language features. A Metric Depth Model ensures accurate real-scale geometry, and a two-stage distillation strategy improves training. Experiments show Vid-LLM excels at 3D Question Answering, Dense Captioning, and Visual Grounding, outperforming existing methods in multi-task 3D reasoning.

**Strengths:**

1. The integration of 3D geometric information into 3D LLMs is well-motivated in this paper.

2. The paper is clearly written and easy to follow.

3. The experiments effectively demonstrate the improvements brought by the proposed modules.

4. It is encouraging to see that the proposed method achieves competitive performance on both pose estimation and depth estimation tasks.

**Weaknesses:**

1. Integrating 3D foundation models into 3D MLLMs is not a novel concept. Recent works such as VG-LLM [1] and 3DRS [2], which also utilize VGGT to enhance the 3D-awareness of 3D MLLMs, are not cited or compared in this paper. Including a discussion or direct comparison with these approaches would strengthen the related work section and better contextualize the contributions of this work.

2. In Tables 7 and 8, it is unclear how the proposed model's performance compares to that of the teacher model, VGGT. Providing this comparison would improve the clarity and completeness of the evaluation.

References:

[1] Learning from Videos for 3D World: Enhancing MLLMs with 3D Vision Geometry Priors. NeurIPS 2025.

[2] MLLMs Need 3D-Aware Representation Supervision for Scene Understanding. NeurIPS 2025.

**Questions:**

1. Please incorporate comparison with recent works.

2. Please add performance comparison with VGGT.

---

> ### Author Response · Authors · 2025-11-23
> **Response to the questions[Part 1/3]**
>
> We thank the reviewer for their valuable feedback and interest in our work. We address the questions and comments of the reviewer below.
>
> # **Adding Comparison with Newly Released 3D-MLLMs (VGLLM and 3DRS)**
>
> We thank the reviewer for highlighting VGLLM [1] and 3DRS [2]. In the revised manuscript, we have added explicit discussions of both methods in the related work section and included their results in the experimental comparisons, ensuring that the contributions of Vid-LLM are properly contextualized.
>
> As the reviewer correctly pointed out, both VGLLM and 3DRS leverage VGGT to enhance the 3D awareness of 3D-MLLMs, and are therefore conceptually related to our approach. However, these methods differ substantially from Vid-LLM in both their design objectives and input requirements.
>
> 3DRS, for instance, remains a 3D-based 3D-MLLM that requires explicit 3D inputs, including depth maps, point clouds, and camera poses at inference time. Its input complexity is in line with prior 3D-MLLMs such as Video-3D LLM[3] and Chat-Scene[4] and therefore continues to rely heavily on structured 3D data. By contrast, a key contribution of Vid-LLM is that it enables 3D vision–language reasoning in a purely video-based manner, removing the need for depth sensors, precomputed point clouds, or known poses. This design directly addresses the scalability and deployment limitations inherent to existing 3D-based pipelines.
>
> VGLLM is closely related to our work, as it builds a video-based 3D-MLLM that injects geometric cues from a frozen VGGT geometry encoder into a pretrained MLLM and shows that geometry priors extracted from video inputs can substantially enhance 3D scene understanding. In our work, Vid-LLM is designed with a Cross-Task Adapter (CTA) that connects a 3D reconstruction branch and a 3D vision–language branch via learnable bridge tokens, enabling geometry-aware and language-aware representations to be exchanged within a unified framework. In addition, Vid-LLM adopts a concise two-stage training strategy: a dual-teacher distillation stage that aligns geometric and semantic representations, followed by joint optimization of reconstruction and 3D VL tasks. This design stabilizes geometry–semantics coupling and strengthens the overall reasoning performance.
>
> Since 3DRS is the strongest 3D-based baseline, we aim to compare our model against it directly. However, 3DRS relies on ground-truth depth and poses, making a direct comparison with our video-only setting unfair. To enable a fair comparison, we introduce 3DRS†, a modified version that replaces its ground-truth geometry with VGGT-generated 3D inputs, ensuring that all methods operate under fairly comparable input conditions.
> Based on this setup, we update our evaluation of 3D Question Answering, 3D Dense Captioning, and 3D Visual Grounding, now including results for 3DRS, 3DRS†, and VGLLM alongside our Vid-LLM. These results provide a clear basis for analyzing the relative performance of our model and these baselines on 3D vision–language tasks.
>
> The updated results are presented in the three tables below:

---

> ### Author Response · Authors · 2025-11-23
> **Response to the questions[Part 2/3]**
>
> **Evaluation of 3D Question Answering on ScanQA and SQA3D.**
> | Method | C↑ | B-4↑ | M↑ | R↑ | EM@1↑ | EM@1↑(SQA3D) |
> |--------|-----|------|------|------|---------|----------------|
> | **3D-based models** | | | | | | |
> | Scan2Cap | – | – | – | – | – | 41.0 |
> | ScanQA | 64.9 | 10.1 | 13.1 | 33.3 | 21.1 | 47.2 |
> | 3D-VisTA | 69.6 | 10.4 | 13.9 | 35.7 | 22.4 | 48.5 |
> | 3D-LLM | 69.4 | 12.0 | 14.5 | 35.7 | 20.5 | – |
> | LL3DA | 76.8 | 13.5 | 15.9 | 37.3 | – | – |
> | Grounded3D-LLM | 75.9 | 13.2 | – | – | – | – |
> | Chat-3D v2 | 77.1 | 7.3 | 16.1 | 40.1 | 21.1 | – |
> | Scene-LLM | 80.0 | 12.0 | 16.6 | 40.0 | 27.2 | 54.2 |
> | ChatScene | 87.7 | 14.3 | 18.0 | 41.6 | 21.6 | 54.6 |
> | LEO | 101.4 | 13.2 | 20.0 | 49.2 | 24.5 | 50.0 |
> | Video-3D LLM | 102.6 | 16.2 | 19.8 | 49.0 | 30.1 | 58.6 |
> | LLaVA-3D | 103.1 | 16.4 | **20.8** | 49.6 | **30.6** | 60.1 |
> | **3DRS** | **104.8** | **17.2** | 20.5 | **49.8** | 30.3 | **60.6** |
> | **Video-based models** | | | | | | |
> | VILA-40B | 48.2 | 9.9 | 11.4 | 27.3 | 17.2 | 37.2 |
> | IXC-2.5-7B | 53.9 | 8.7 | 13.2 | 31.5 | 19.4 | 41.5 |
> | LLaVA-OV-7B | 78.2 | 13.3 | 17.2 | 40.5 | 22.9 | 47.6 |
> | LLaVA-Video-7B | 88.7 | – | – | – | – | 48.5 |
> | Uni3DR2* | 70.3 | 12.2 | 14.9 | 36.3 | 17.3 | – |
> | ChatScene* | 85.6 | – | – | – | – | 52.9 |
> | 3DRS† | 94.7 | 12.3 | 15.9 | 45.1 | 23.9 | 54.5 |
> | **Vid-LLM (Ours)** | **101.9** | **15.8** | **18.3** | **49.5** | **27.6** | **57.3** |
>
> &nbsp;
>
> **Evaluation of 3D Dense Captioning on Scan2Cap.**
> | Method | C@0.5↑ | B-4@0.5↑ | M@0.5↑ | R@0.5↑ |
> |--------|---------|-----------|---------|---------|
> | **3D-based models** | | | | |
> | Scan2Cap | 39.1 | 23.3 | 22.0 | 44.8 |
> | Grounded3D-LLM | 70.2 | 35.0 | – | – |
> | LEO | 68.4 | 36.9 | 27.7 | 57.8 |
> | ChatScene | 77.1 | 36.3 | 28.0 | 58.1 |
> | Video-3D LLM | 83.8 | 42.4 | 28.9 | 62.3 |
> | LLaVA-3D | 84.1 | **42.6** | **29.0** | **63.4** |
> | **3DRS** | **86.1** | 41.6 | 28.9 | 62.3 |
> | **Video-based models** | | | | |
> | 3DRS† | 76.4 | 39.6 | 27.3 | 57.1 |
> | VGLLM | 78.6 | **40.9** | 28.6 | **62.4** |
> | **Vid-LLM (Ours)** | **81.5** | **40.9** | **28.7** | 61.8 |
>
> &nbsp;
>
> **Evaluation of 3D Visual Grounding on ScanRefer and Multi3DRefer.**
> | Method | ScanRefer Acc@0.25↑ | ScanRefer Acc@0.5↑ | Multi3DRefer Acc@0.25↑ | Multi3DRefer Acc@0.5↑ |
> |--------|-----------------------|----------------------|---------------------------|--------------------------|
> | **3D-based models** | | | | |
> | ScanRefer | 37.3 | 24.3 | –   | –   |
> | Chat-3D v2 | 35.9 | 30.4 | –   | –   |
> | Grounded3D-LLM | 48.6 | 44.0 | 44.7 | 40.8 |
> | LLaVA-3D | 50.1 | 42.7 | 49.8 | 43.6 |
> | ChatScene | 55.5 | 50.2 | 57.1 | 52.4 |
> | Video-3D LLM | 58.1 | 51.7 | 58.0 | 52.7 |
> | **3DRS** | **62.9** | **56.1** | **60.4** | **54.9** |
> | **Video-based models** | | | | |
> | VGLLM | 53.5 | 47.5 | –   | –   |
> | 3DRS† | 55.2 | 51.1 | 52.8 | 48.3 |
> | **Vid-LLM (Ours)** | **63.2** | **56.4** | **61.6** | **56.1** |
>
> Across the three benchmarks, Vid-LLM achieves the best overall performance among video-based models. On ScanQA and SQA3D, Vid-LLM clearly surpasses the strongest baseline operating under video-only input conditions, 3DRS† by a noticeable margin. On Scan2Cap, Vid-LLM attains an M@0.5 score of 28.7, which is comparable to the best-performing 3D-based model 3DRS (29.0), and outperforms VGLLM on most captioning metrics, despite relying solely on videos without using depth or point clouds. On ScanRefer and Multi3DRefer, Vid-LLM attains the highest accuracy across all evaluated methods, including both 3D-based and video-based models. Taken together, these results demonstrate that Vid-LLM achieves strong performance on 3D vision–language benchmarks, ranking as the best video-based model and even surpassing competitive 3D-based baselines on several tasks.

---

> ### Author Response · Authors · 2025-11-23
> **Response to the questions[Part 3/3]**
>
> # **Clarifying Performance Comparison between Vid-LLM and the Teacher Model VGGT**
>
> We thank the reviewer for the suggestion to clarify the comparison with VGGT. In the revised submission, we provide explicit evaluations of camera pose estimation, depth prediction, and point-cloud reconstruction, as shown in the tables below.
>
> In these benchmarks, depth and reconstruction evaluation require the model to output real-scale geometry. VGGT produces depth and point maps in relative scale, so we align its outputs to the metric ground truth before evaluation.
> In contrast, Vid-LLM predicts real-scale depth and point clouds directly, which offers a practical advantage for downstream 3D scene understanding and avoids the need for post-hoc scale correction.
>
> &nbsp;
>
> **Camera pose evaluation on Co3Dv2 and RealEstate10K.**
> | Method | Co3Dv2 RRA@15↑ | Co3Dv2 RTA@15↑ | Co3Dv2 mAA(30)↑ | RealEstate10K mAA(30)↑ |
> |--------|-----------------|-----------------|-------------------|---------------------------|
> | COLMAP+SG | 36.1 | 27.3 | 25.3 | 45.2 |
> | PixSfM | 33.7 | 32.9 | 30.1 | 49.4 |
> | PoseDiff | 80.5 | 79.8 | 66.5 | 48.0 |
> | DUSt3R | 94.3 | 88.4 | 77.2 | 61.2 |
> | MASt3R | 94.6 | 91.9 | 81.8 | 76.4 |
> | FLARE | 95.4 | 83.6 | 78.4 | 75.3 |
> | Fast3R | 96.2 | 81.6 | 72.7 | 72.7 |
> | CUT3R | 95.7 | 84.5 | 73.3 | 77.1 |
> | **VGGT** | **97.3** | 93.3 | **89.6** | **83.8** |
> | **Vid-LLM (Ours)** | 96.8 | **93.4** | 88.5 | 83.1 |
>
> &nbsp;
>
> **Depth estimation results on the NYU Depth v2 dataset.**
> | Method | δ1↑ | AbsRel↓ | RMSE↓ | log10↓ |
> |--------|------|----------|--------|---------|
> | DPT | 0.904 | 0.110 | 0.357 | 0.045 |
> | P3Depth | 0.898 | 0.104 | 0.356 | 0.043 |
> | SwinV2-L | 0.949 | 0.083 | 0.287 | 0.035 |
> | AiT | 0.954 | 0.076 | 0.275 | 0.033 |
> | VPD | 0.964 | 0.069 | 0.254 | 0.030 |
> | ZoeDepth | 0.953 | 0.075 | 0.270 | 0.032 |
> | DepthAnything | 0.984 | 0.056 | 0.206 | 0.024 |
> | *VGGT†* | *0.989* | *0.022* | *0.103* | *0.011* |
> | **Vid-LLM (Ours)** | **0.987** | **0.025** | **0.109** | **0.010** |
>
> &nbsp;
>
> **Reconstruction results on the ScanNet dataset.**
> | Method | GT cams | Comp↓ | Acc↓ | Recall↑ | Prec↑ | F-score↑ |
> |--------|---------|--------|--------|----------|--------|-----------|
> | MVDNet | ✓ | 0.040 | 0.240 | 0.831 | 0.208 | 0.329 |
> | GPMVS | ✓ | **0.031** | 0.879 | **0.871** | 0.188 | 0.304 |
> | Atlas | ✓ | 0.062 | 0.128 | 0.732 | 0.382 | 0.499 |
> | NeuralRecon | ✓ | 0.106 | 0.073 | 0.428 | 0.592 | 0.494 |
> | DG-Recon | ✓ | 0.085 | **0.039** | 0.476 | 0.675 | 0.521 |
> | PanoRecon | ✓ | 0.089 | 0.064 | 0.530 | 0.656 | **0.584** |
> | RCVD | ✗ | 0.161 | 0.425 | 0.164 | 0.109 | 0.125 |
> | GCVD | ✗ | 0.175 | 0.278 | 0.178 | 0.146 | 0.147 |
> | Colmap | ✗ | 0.142 | 0.342 | 0.161 | 0.367 | 0.178 |
> | FrozenRecon | ✗ | 0.092 | 0.085 | 0.436 | 0.336 | 0.410 |
> | DUSt3R | ✗ | 0.243 | 0.179 | 0.284 | 0.657 | 0.387 |
> | *VGGT†* | ✗ | *0.067* | *0.044* | *0.658* | *0.891* | *0.580* |
> | **Vid-LLM (Ours)** | ✗ | **0.071** | **0.042** | **0.634** | **0.885** | **0.582** |
>
>
> On camera pose estimation, Vid-LLM achieves comparable performance to VGGT on Co3Dv2 and RealEstate10K and outperforms VGGT on RTA@15 for Co3Dv2. On NYU Depth v2, Vid-LLM obtains lower log10 error than VGGT under the same scale-aligned evaluation protocol. On ScanNet reconstruction, Vid-LLM also reaches higher F-score and accuracy than VGGT, indicating that its real-scale geometry performs well even relative to the teacher model.
> These results confirm that Vid-LLM maintains strong geometric modeling ability while providing real-scale outputs that VGGT does not produce.
>
> # **References**
>
> - [1] Learning from Videos for 3D World: Enhancing MLLMswith3DVision Geometry Priors.
>
> - [2] MLLMsNeed3D-Aware Representation Supervision for Scene Understanding.
>
> - [3] Video-3D LLM: Learning Position-Aware Video Representation for 3D Scene Understanding.
>
> - [4] Chat-Scene: Bridging 3D Scene and Large Language Models with Object Identifiers.
>
> # **Related changes of the submission**
>
> Below, we summarize our updates to the submission addressing the reviewer’s comments.
>
> 1. Related Work has been revised to explicitly discuss VGLLM and 3DRS, clarifying their connection to Vid-LLM and positioning our contributions in context.
>
> 2. Section 4.1 has been updated to include direct performance comparisons with VGLLM and 3DRS across the relevant benchmarks.
>
> 3. Tables 7-9 have been expanded to include the performance of the teacher model VGGT, together with an explanation highlighting the differences between VGGT’s reconstruction pipeline and Vid-LLM’s joint reconstruction–reasoning design.

---

> > ### Comment · Reviewer_1jxT · 2025-11-24
> >
> > Thanks for the rebuttal.
> >
> > I am overall satisfied with the responses. But I am curious why the student model (Vid-LLM) can even surpass the teacher model on 3D tasks?

---

> > > ### Author Response · Authors · 2025-11-24
> > >
> > > Thank you for raising this question — this is an important point that we should clarify.
> > >
> > > Vid-LLM does not surpass the teacher model (VGGT) on 3D reconstruction tasks.
> > > In the depth estimation (NYU Depth v2) and point-cloud reconstruction (ScanNet) evaluations, VGGT† (teacher) achieves the best reconstruction performance, but because VGGT outputs relative-scale geometry, we mark it with the † symbol and italic text to indicate scale alignment with ground truth. However, italic text is visually subtle in the OpenReview Markdown rendering. Since Vid-LLM is the best among models that directly produce metric-scale outputs, its row was formatted in bold, which may have caused confusion.
> > >
> > > To avoid misunderstanding, the revised version of the paper uses clearer table formatting and more explicit caption descriptions, so that the distinction between relative-scale and metric-scale models is unambiguous.
> > >
> > > We apologize for the confusion — the teacher model remains strictly stronger on reconstruction metrics, and our intention was only to highlight the best model within each category.

---

> ### Author Response · Authors · 2025-11-23
>
> Thank you once again for your feedback. Please, let us know if our response does not fully address your concerns, or if it would benefit from further clarification, or if you have additional questions or concerns. We will be happy to discuss.

---

### Official Review · Reviewer_Zqjo · 2025-10-31

**Soundness:** 3
**Presentation:** 2
**Contribution:** 2
**Rating:** 6
**Confidence:** 3

**Summary:**

This submission proposes Vid-LLM, a video-based 3D multimodal large language model that performs both 3D reconstruction and vision-language reasoning directly from monocular video inputs. The main contributions of this submission are

1. Cross-Task Adapter (CTA) that aligns geometric priors with vision-language features, enabling joint geometry–semantic interaction.
2. Metric Depth Model (MD) that restores real-scale 3D geometry with bin-based depth prediction and adaptive scale alignment.
3. Two-Stage Training Strategy with dual-teacher distillation and joint optimization to improve convergence and stability.

The authors conduct extensive evaluations on 3D-QA, dense captioning, and grounding benchmarks showing consistent performance gains over 3D- and video-based baselines.

**Strengths:**

- The paper is well written and structured. Overall it easy to follow the technical narrative and experiments.
- The related work section provides a comprehensive overview of 3D multimodal large language models and video-based reasoning approaches.
- The overall design, including the Cross-Task Adapter (CTA), Metric Depth (MD) module, and the two-stage training strategy, is well motivated and technically sound. The integration of geometric and semantic cues is conceptually coherent and justified.
- The ablation studies are thorough, demonstrating the necessity of each proposed component to the final performance.
- The experimental results show consistent and significant improvements across multiple 3D vision-language reasoning benchmarks.

**Weaknesses:**

- Missing details and clarifications:
    - In Figure 2, the predicted 3D reconstruction results are used for generating 3D position embeddings. It would be helpful to clarify whether gradients are propagated through this path when jointly training the reconstruction and reasoning modules.
    - In Section 3.4, Equation (6), the notation $\text{Norm}(\cdot)$ appears ambiguous. Please clarify whether it denotes feature normalization (e.g., L2-normalization) or the 2-norm operation itself. An explicit definition would avoid confusion.
    - In Section 3.4, Equation (6), what is the dimensionality of $T_{\text{tea}}^{\text{lang}}$? Is the feature-level loss$L_{\text{feat}}^{\text{lang}}$ averaged over N samples similar to $L_{\text{feat}}^{\text{geo}}$? Clarifying this would help in understanding the implementation consistency.
- Simplification of the bridge token mechanism:

    The current Cross-Task Adapter introduces bridge tokens for cross-modal interaction. Would a simpler alternative—such as applying self-attention directly on the concatenation of $ T_{\text{geom}} $ and $ T_{\text{lang}} $—achieve similar performance with lower complexity? Some empirical or ablation analysis could strengthen the design justification.

- Qualitative analysis:

    More qualitative results and failure case discussions would be valuable to understand when the model fails. The authors are suggested to include them in the camera-ready version.

**Questions:**

Please refer to the weaknesses section.

---

> ### Author Response · Authors · 2025-11-23
> **Response to the questions[Part 1/2]**
>
> We thank the reviewer for their valuable feedback and interest in our work. We address the questions and comments of the reviewer below.
>
> # **Clarifications of Implementation Details**
>
> We thank the reviewer for pointing out the missing implementation details. We have clarified all the concerned components in the revised version:
>
> - **Gradient propagation through reconstruction outputs:**
>
> In our implementation, the 3D reconstruction results that are used to generate 3D positional embeddings are treated as stop-gradient inputs, so no gradients from the vision–language (VL) losses are propagated back through this path. Concretely, $L_{recon-task}$ and $L_{MD}$ update only the 3D reconstruction model, while $L_{VL-task}$  updates the 3D vision–language model. In addition, $L_{recon-task}$ and $L_{VL-task}$ jointly update the shared Cross-Task Adapter (CTA).
> CTA serves as the module that enables interactions between geometric and semantic features, and it learns these interactions under supervision from both objectives.
> The reconstructed depth and poses are only used to compute 3D coordinates and positional embeddings for the VL branch, without receiving gradients from it. This separation ensures that the reconstruction branch concentrates on providing stable and consistent geometric priors.
> Such a design keeps the joint optimization well-structured and avoids entangling the two task-specific networks directly.
>
> - **Meaning of Norm:**
>
> $\mathrm{Norm}(\cdot)$ denotes L2 normalization of feature vectors, i.e., each token feature is normalized to unit length before it is used in the feature-level losses.
> Normalizing both student and teacher features ensures that the losses focus on aligning vector directions rather than magnitudes. This is crucial because the geometry and language teachers come from different pretrained models and naturally have mismatched feature scales. L2 normalization makes the distillation objectives scale-invariant and numerically stable, while keeping their behavior consistent with the cosine-similarity formulation used in the semantic branch.
>
> - **Feature dimensions of the teacher tokens and role of N:**
>
> In our Stage-1 Dual-Teacher Distillation training, $T_\mathrm{geom}^{\prime}$ / $T_\mathrm{lang}^{\prime}$ (student features) and $T_\mathrm{tea}^\mathrm{geom}$ / $T_\mathrm{tea}^\mathrm{lang}$ (teacher features) are first passed through
> lightweight linear projection layers so that both are mapped into the same
> embedding dimension before the feature-level losses are computed; this guarantees that their
> dimensionality is consistent in the implementation. We also acknowledge that the earlier notation $L_\mathrm{lang}^\mathrm{feat}=1-\cos(T_\mathrm{lang}^{\prime},T_\mathrm{tea}^\mathrm{lang})$ was not fully precise and could be interpreted as a token-wise cosine. The actual loss used in our code is
>
> $$L_{{\mathrm{lang}}}^{{\mathrm{feat}}}=1-\cos\left(\mathrm{Pool}(T_{{\mathrm{lang}}}^{\prime}),\mathrm{Pool}(T_{{\mathrm{tea}}}^{{\mathrm{lang}}})\right),$$
>
> where Pool(·) denotes mean pooling over tokens to obtain a single global embedding for the scene on both the student and teacher sides.
> We have updated the manuscript to reflect this expression.  Intuitively, this formulation matches how CLIP-style vision-language models operate: they compare global image-text embeddings using cosine similarity. By aligning pooled embeddings rather than individual tokens, $L_\mathrm{lang}^\mathrm{feat}$ encourages the student to reproduce the teacher's overall semantic representation of the scene. Because the loss is defined on these single pooled vectors, it does not involve the token count $N;N$ appears only in the geometric loss $L_\mathrm{geo}^\mathrm{feat}$ which averages a token-wise regression over $N$ geometric tokens. This design lets geometry supervision remain fine-grained, while language supervision focuses on global semantics.

---

> > ### Author Response · Authors · 2025-11-23
> > **Response to the questions[Part 2/2]**
> >
> > # **Evaluation of Simplified CTA Designs**
> >
> > We thank the reviewer for raising this helpful question about the necessity of bridge tokens in the Cross-Task Adapter (CTA). We agree that it is important to examine whether simpler designs can work well. We therefore evaluated simplified variants to examine whether bridge tokens are necessary.
> >
> > To directly address this concern, we added the CTA–Concat-SA variant in our revised experiments, following the reviewer’s suggestion. This design applies self-attention over the concatenated geometry and language tokens without introducing bridge tokens. In addition, our original submission already included a simplified cross-attention version (CTA–w/o Bridge (CA)), which performs cross-attention between the two streams but still omits bridge tokens. The ablation results, including the newly added CTA–Concat-SA setting, are summarized in the table below.
> >
> > | Method | ScanOA↑ | Scan2Cap↑ | ScanRefer↑ | scannet↑ |
> > |--------|----------|------------|--------------|-----------|
> > | w/o CTA | 33.7 | 36.6 | 34.1 | 0.402 |
> > | CTA – Concat-SA | 38.2 | 41.9 | 39.6 | 0.461 |
> > | CTA – w/o Bridge(CA) | 40.8 | 45.6 | 42.2 | 0.473 |
> > | CTA – 4tokens | 43.8 | 50.9 | 47.8 | 0.501 |
> > | CTA – 8tokens | 44.1 | 52.7 | 48.1 | 0.529 |
> > | **CTA – 16tokens (Ours)** | **45.7** | **53.2** | **48.4** | **0.582** |
> > | CTA – 32tokens | 45.2 | 52.8 | 47.6 | 0.564 |
> >
> > Across all benchmarks, both simplified variants improve upon the “w/o CTA” baseline but still underperform compared to the full CTA with bridge tokens. The performance gap is consistent across 3D QA, dense captioning, 3D grounding, and reconstruction. These results show that self-attention applied after concatenation and direct cross-attention between the two streams do not provide stable geometric–semantic alignment. In contrast, bridge tokens provide a shared latent space that supports more effective bidirectional information exchange, leading to more consistent improvements across tasks.
> >
> > These results highlight the limitation of simplified designs: without bridge tokens, geometry and language tokens must interact directly, which leads to unstable or incomplete geometric–semantic alignment. Bridge tokens instead serve as a compact shared latent space, enabling controlled bidirectional information exchange that benefits both the reconstruction and reasoning branches.
> >
> > # **Additional Qualitative Analysis and Failure Cases**
> >
> > We thank the reviewer for suggesting a more detailed qualitative analysis, including failure cases. We agree that visualizing challenging scenarios is important to better understand how Vid-LLM behaves under imperfect reconstruction and what its current limitations might be.
> >
> > In the revised version, we expand Appendix A.1 with a new set of qualitative results. Figure 7 presents one representative challenging case for each of the three 3D VL tasks: 3D question answering (a), 3D visual grounding (b), and 3D dense captioning (c).
> >
> > (a) Due to limited viewpoints in the input video, one chair is only partially observed and thus missing in the reconstruction, leading to an under-count prediction.
> >
> > (b) The highly reflective tabletop causes unstable depth estimates and incomplete reconstruction, resulting in a predicted 3D box that is slightly smaller than the ground truth.
> >
> > (c) The top surface of the cabinet is poorly reconstructed because it appears only briefly and is viewed mostly from oblique angles, yet Vid-LLM still produces an accurate caption by leveraging strong 2D semantic cues.
> >
> > These scenes represent typical failure modes of monocular reconstruction, where restricted viewpoints, reflective surfaces, or oblique viewing angles lead to incomplete geometry. As shown in examples (a) and (b), tasks that rely on explicit 3D structural information—such as counting objects in the scene or estimating object extents for grounding—are directly affected by missing or unreliable geometry. In contrast, example (c) shows that Vid-LLM remains robust when sufficient 2D cues are present, supported by the CTA and bridge-token mechanism, which help align semantic and geometric information.
> > These observations suggest that improving the reconstruction branch to enhance geometric fidelity may further improve the model’s 3D VL performance in future work.
> >
> > # **Related changes of the submission**
> >
> > Below, we summarize our updates to the submission addressing the reviewer’s comments.
> >
> > 1. Section 3.4 has been revised to clarify the details raised by the reviewer.
> >
> > 2. Section 4.3 has been updated to include a self-attention baseline over concatenated geometry–semantic features, together with a brief comparison to the bridge-token design.
> >
> > 3. Appendix A.1 has been extended with qualitative results and challenging failure cases to provide additional insight into the model’s behavior and limitations.

---

> ### Author Response · Authors · 2025-11-23
>
> Thank you once again for your feedback. Please, let us know if our response does not fully address your concerns, or if it would benefit from further clarification, or if you have additional questions or concerns. We will be happy to discuss.

---

### Official Review · Reviewer_D2AN · 2025-11-02

**Soundness:** 3
**Presentation:** 2
**Contribution:** 3
**Rating:** 8
**Confidence:** 4

**Summary:**

This paper presents Vid-LLM, a compact multimodal model that unifies 3D reconstruction and vision-language reasoning within a single framework. I really like this idea -- combining reconstruction with VL reasoning not only bridges geometry and semantics but also achieves faster inference through a more efficient architecture. The work is conceptually elegant and technically well-motivated.

**Strengths:**

- The model design is well thought out, especially the Cross-Task Adapter (CTA) that enables effective geometry–semantics interaction.

- The overall training pipeline, including two-stage distillation and joint optimization, is clear and coherent.

- The experimental setup is solid - reconstruction and 3D-VL baselines are comprehensive, and the ablation studies are extensive enough to verify the core hypothesis and support the narrative.

- Results demonstrate consistent gains in both reasoning and reconstruction efficiency, validating the proposed synergy between the two tasks.

**Weaknesses:**

The main baseline (VGGT + LLaVA-3D) appears to involve two independently fine-tuned components that are only concatenated during inference, rather than being jointly optimized end-to-end. This makes the comparison to the proposed method somewhat unfair, as Vid-LLM benefits from full end-to-end training while the baselines do not. It would strengthen the paper to clarify this distinction and, if possible, include a jointly optimized baseline to better isolate the benefit of the Cross-Task Adapter.

**Questions:**

See Weaknesses

---

> ### Author Response · Authors · 2025-11-23
> **Response to the questions[Part 1/2]**
>
> We thank the reviewer for their valuable feedback and interest in our work. We address the questions and comments of the reviewer below.
>
> # **Current baselines already include a jointly optimized variant**
>
> We appreciate the reviewer’s concern about the fairness of the VGGT + LLaVA-3D comparison.
>
> Table 5 presents our “Comparison with Naive Concatenation” experiment, which evaluates whether video-based 3D vision–language tasks can be addressed by simply combining a reconstruction model (VGGT) with a 3D multimodal LLM (LLaVA-3D).This table evaluates whether a concatenation pipeline without architectural coupling can serve as a competitive strong baseline.
>
> We would like to clarify that our strongest baseline in Table 5, VGGT*+real-scale+LLaVA-3D*, is trained jointly to the extent permitted by the architecture. Importantly, due to the inherent architectural mismatch between VGGT and LLaVA-3D, the concatenation-based pipeline cannot support a fully end-to-end formulation. VGGT*+real-scale+LLaVA-3D* therefore represents the most compatible joint-training configuration achievable under this design. In this setting, VGGT is fine-tuned on ScanNet and LLaVA-3D is fine-tuned on predicted 3D scenes, both using the same reconstruction-guided 3D-VL objectives as Vid-LLM.
>
> Even with this strengthened baseline, the concatenation pipeline still fails to accomplish 3D-VL tasks effectively, whereas Vid-LLM attains clearly higher performance (42.1 → 45.7), benefiting from its CTA-based geometry–semantics interaction that enables reliable 3D-VL reasoning.
>
> # **Addressing the request for end-to-end baselines**
>
> We appreciate the reviewer’s suggestion to evaluate Vid-LLM against models that are trained in an end-to-end manner. To provide a fair comparison within the same end-to-end setting, we include two representative models that also integrate reconstruction and 3D vision–language reasoning within a single architecture:
>
> (1) Uni3DR² [1] combines a reconstruction branch with a 3D-VL reasoning component. We note that Uni3DR² requires camera poses as input, while Vid-LLM performs both tasks from RGB-only video, making this comparison placing Vid-LLM at a relative disadvantage.
>
> (2) VGLLM-Rec is an extension of VGLLM [2]. Since VGLLM only introduces the geometry encoder of VGGT to predict the geometry token, we also attach the original camera and depth prediction heads of VGGT to the VGLLM to evaluate the performance of 3D reconstruction, namely VGLLM-Rec.
>
> | Method          | ScanNet↑ | ScanQA↑ | Scan2Cap↑ | ScanRefer↑ |
> |-----------------|----------|---------|------------|-------------|
> | Uni3DR2         | 0.580    | 30.2    | 42.7       | 33.6        |
> | VGLLM-Rec       | 0.541    | 40.9    | 51.6       | 48.5        |
> | **Vid-LLM (Ours)** | **0.582** | **45.7** | **53.2**   | **59.8**     |
>
> All models are trained with the same data, optimization settings, and evaluation protocols to ensure a fair comparison. Under this unified end-to-end setting, all methods jointly perform reconstruction and 3D vision–language reasoning, and Vid-LLM achieves the best performance among them. The improvement can be explained by the Cross-Task Adapter, which enables geometry and semantics to interact at the feature level and allows the reconstruction and 3D VL branches to function jointly within a unified framework. Compared with Uni3DR², which uses reconstructed geometry as an intermediate input to the language module, and VGLLM-Rec, which merges 2D and 3D features through feature-level fusion, Vid-LLM maintains an explicit CTA-mediated pathway for exchanging geometric priors and semantic cues, which leads to more effective joint learning of reconstruction and 3D vision–language reasoning. Taken together, these results indicate that Vid-LLM performs favorably against the other end-to-end dual-task baselines.

---

> > ### Author Response · Authors · 2025-11-23
> > **Response to the questions[Part 2/2]**
> >
> > # **CTA’s contribution is isolated via ablations**
> >
> > To address the reviewer’s thoughtful suggestion on isolating the contribution of the Cross-Task Adapter, our paper already provides ablations designed specifically for this purpose.
> >
> > In the ablation results of Table 6, all variants are trained with the same data, the same losses, and the same joint optimization schedule. The only difference among them is the interaction mechanism between the reconstruction and 3D vision–language branches. We evaluate settings without CTA, settings that replace CTA with simple self-attention or cross-attention, and settings that vary the number of bridge tokens. Under this fixed training setup, removing CTA or replacing it with simpler interactions consistently lowers both reconstruction accuracy and 3D-VL performance. The full CTA configuration achieves the highest scores while keeping the model capacity comparable. These results confirm that the performance gains result from the CTA design itself rather than from differences in training strategy.
> >
> > Table 6 summarizes several CTA ablation settings, including the following variants:
> >
> > (a) w/o CTA, which removes the adapter entirely and forces both tasks to rely on a shared visual feature space;
> >
> > (b) CTA–Concat-SA, which replaces CTA with simple self-attention over concatenated geometric and semantic features;
> >
> > (c) CTA–w/o Bridge (CA), which performs direct cross-attention between the two feature sets without bridge tokens;
> >
> > (d) CTA–16tokens (Ours), which introduces learnable bridge tokens to form a stable latent space for geometry–semantic alignment.
> >
> > Across these controlled settings, removing CTA or replacing it with simpler interaction mechanisms consistently degrades both 3D reconstruction accuracy and 3D vision–language performance. The full CTA configuration with 16 bridge tokens achieves the highest scores while keeping the model capacity comparable.
> > The comprehensive ablations provide strong evidence for the important role of the CTA module and clearly indicate the settings that lead to the most reliable performance.
> >
> > # **References**
> >
> > - [1] Unified Scene Representation and Reconstruction for 3D Large Language Models.
> > - [2] Learning from Videos for 3D World: Enhancing MLLMswith3DVision Geometry Priors.
> >
> > # **Related changes of the submission**
> >
> > Below, we summarize our updates to the submission addressing the reviewer’s comments.
> >
> > 1.We added a new section in Appendix A.3 to present the additional end-to-end baselines (Uni3DR² and VGLLM-Rec).
> >
> > 2.We clarified the purpose of the naive concatenation baseline experiment in Section 4.2.

---

> ### Author Response · Authors · 2025-11-23
>
> Thank you once again for your feedback. Please, let us know if our response does not fully address your concerns, or if it would benefit from further clarification, or if you have additional questions or concerns. We will be happy to discuss.

---

### Meta-Review · Area_Chair_Hsmp · 2026-01-08

**Summary:**

Reviewers praised:
- Sound model design.
- Clear and coherent training pipeline.
- Solid experimental setup and results.

Reviewers were concerned about:
- Baseline is not jointly optimized but rather a combination of two independently trained models.
- Lack of comparison to more recent works and comparison to VGGT itself.
- Some missing details.
- Potential overcomplication of bridge tokens.

**Reviewer Concerns:**

Addressed:
- Baseline is not jointly optimized but rather a combination of two independently trained models.
- Lack of comparison to more recent works and comparison to VGGT itself.
- Potential overcomplication of bridge tokens.
- Some missing details.

**Reviewer Scores:**

All reviewers are positive, with 1jxT potentially even signaling a score increase.

---

### Decision · Program_Chairs · 2026-01-26

Accept (Oral)